# Liver Dangers of Herbal Products: A Case Report of Ashwagandha-Induced Liver Injury

**DOI:** 10.3390/ijerph20053921

**Published:** 2023-02-22

**Authors:** Marta Lubarska, Przemysław Hałasiński, Szymon Hryhorowicz, Dagmara Santabye Mahadea, Liliana Łykowska-Szuber, Piotr Eder, Agnieszka Dobrowolska, Iwona Krela-Kaźmierczak

**Affiliations:** 1Department of Gastroenterology, Dietetics and Internal Diseases, University of Medical Sciences Poznan, Przybyszewskiego 49, 60-355 Poznan, Poland; 2Institute of Human Genetics, Polish Academy of Sciences, Strzeszynska 32, 60-479 Poznan, Poland

**Keywords:** liver injury, toxin, ashwagandha, herbal supplement

## Abstract

In recent years, cases of liver damage caused by ashwagandha herbal supplements have been reported from different parts of the world (Japan, Iceland, India, and the USA). Here, we describe the clinical phenotype of suspected ashwagandha-induced liver injury and the potential causative mechanism. The patient was admitted to the hospital because of jaundice. In the interview, it was reported that he had been taking ashwagandha for a year. Laboratory results showed an increase in total bilirubin, alanine transaminase (ALT), aspartate transaminase (AST), (gamma-glutamyl transpherase (GGT), alkaline phosphatase (ALP), total cholesterol, triglycerides, and ferritin. Based on clinical symptoms and additional tests, the patient was diagnosed with acute hepatitis and referred to a facility with a higher reference rate to exclude drug-induced liver injury. An R-value was assessed, indicative of hepatocellular injury. The result of the 24 h urine collection exceeded the upper limit of normal for copper excretion in urine twice. The clinical condition improved after intensive pharmacological treatment and four plasmapheresis treatments. This case is another showing the hepatotoxic potential of ashwagandha to cause cholestatic liver damage mixed with severe jaundice. In view of several documented cases of liver damage caused by ashwagandha and the unknown metabolic molecular mechanisms of substances contained in it, attention should be paid to patients reporting the use of these products in the past and presenting symptoms of liver damage.

## 1. Introduction

The use of herbs and dietary supplements (HDS) is increasing worldwide. Their association with liver damage has become a focus of interest, especially due to the increasing rates of hepatotoxicity caused by their use. The regulation of HDS use in the U.S. is strictly defined by law, while regulations in Europe do not specify the need to test the safety of HDS before marketing them [1,2]. The true incidence of HDS-induced hepatotoxicity is, therefore, not fully known. Although HDS includes many products sold for a variety of uses, they can be divided into several main groups. One is herbal products, including traditional botanicals such as Chinese or Ayurvedic herbs. Suspected hepatotoxic ingredients include green tea extract, obtained from the Chinese tea tree *Camellia sinensis*, which consists of catechins and polyphenolic flavanols. Depending on the dose used, it can cause liver damage by damaging mitochondria and creating reactive oxygen species, thereby inducing oxidative stress and apoptosis of liver cells [3]. Another example is *Garcinia cambogia*, also known as Malabar tamarind, which contains the active ingredient hydroxy citric acid. In studies, it has been linked to cases of liver damage through liver fibrosis, inflammation, and the induction of oxidative stress [4]. Another example of a naturally occurring compound is usnic acid, present in several lichen species, which, despite its antibacterial, antiviral, and anti-inflammatory properties [5,6], has caused hepatotoxicity, most likely due to mitochondrial dysfunction and oxidative stress [7].

More examples of herbal products associated with liver damage caused by mitochondrial oxidative stress are: *Ephedra sinica*, known as Chinese ephedra or *ma huang* which, due to its alkaloid compounds, acts like ephedrine and pseudoephedrine [8] *Kava Kava* (*Piper methysticum* root), being a traditional medicinal product due to its anxiolytic properties [9,10], and *Withania somnifera* root (ashwagandha), despite conflicting reports.

Ashwagandha extracts contain more than 200 primary and secondary metabolites, including alkaloids, flavone glycosides, glycovitanolides, steroidal lactones (withanolides), saponins, sterols, and phenols [11] that can be used for the prevention and treatment of various diseases such as arthritis, obesity, anxiety, hypothyroidism, cognitive and neurological disorders, and Parkinson disease [12,13,14,15]. To date, the literature has reported more than 12 alkaloids and 35 withanolides that have been purified, identified, and characterized from the roots, fruits, leaves, and aerial parts of *Withania* species [11,16,17]. It has been reported that these extracts have antioxidant, neuroprotective, immunomodulatory, anticancer, and antiviral properties [18,19,20,21]. Ashwagandha, like most herbal supplements, is generally considered safe to consume without significant side effects.

Studies have shown the beneficial effects of *Withania somnifera* on inflammation by stimulating lymphocyte and NK cell proliferation [22], as well as by inhibiting the reaction of monovalent copper ions with hydrogen peroxide, thereby contributing to the inhibition of copper-induced oxidative stress, consequently alleviating inflammation [23]. A murine model study demonstrated the hepatoprotective effects of ashwagandha during gentamicin treatment, but caution should be taken in extrapolating these results to the human population [24]. However, with the increased use of herbal medicines and dietary supplements containing ashwagandha, induced liver injury and liver failure are becoming more common [25,26,27,28,29,30].

## 2. Case Presentation

A 23-year-old Polish male without any medical history was admitted to the Department of Gastroenterology, Dietetics and Internal Diseases of Heliodor Święcicki University Hospital in Poznan, Poland in October 2019 due to jaundice that appeared a few days prior. The patient reported pruritus, malaise, fatigue, gastrointestinal disorders, and stool discoloration. The patient did not take any medications regularly. He had been recently consuming ashwagandha dietary supplements (the dose and quality of the supplement are unknown). Serological tests for viral hepatitis were ordered and viral etiology was excluded (hepatitis A virus—HAV; hepatitis E virus—HEV; hepatitis B virus—HBV; hepatitis C virus—HCV and human immunodeficiency virus—HIV). Additionally, laboratory tests were performed and showed elevated total bilirubin (11.5 mg/dL), ALT (alanine transaminase) (490.0 U/L), AST (aspartate transaminase) (234.0 U/L), GGT (gamma-glutamyl transpherase) (99 U/L), ALP (alkaline phosphatase) (227 U/L), total cholesterol (278 mg/dL), triglycerides (378 mg/dL), and ferritin (971 µg/L) levels. The patient was diagnosed with acute hepatitis. Additionally, genetic tests for hemochromatosis were performed, and the result was found to be negative. No evidence of a Kayser–Fleischer ring was found upon an ophthalmological evaluation. Abdominal MRI with contrast showed no features of focal liver lesions. In the magnetic resonance cholangiopancreatography (MRCP) images, the intra- and extrahepatic bile ducts were not dilated (common hepatic duct—CHD and common biliary duct—CBD, 4 mm in diameter, drain into the duodenum through the ampulla of Vater). During hospitalization, a further increase in the concentration of bilirubin, and thus a significant intensification of pruritus, was observed. Treatment with methylprednisone, ornithine aspartate, ursodeoxycholic acid, potassium, hyoscine, and lactulose was introduced. The patient was referred to a higher-reference hospital for further diagnosis. Tests to diagnose autoimmune hepatitis were performed. ANA and SMA antibodies were not detected, and IgG concentration was normal. In the absence of AMA antibodies, primary cholangitis (PBC) was also excluded. MRCP and ERCP imaging were found to be normal, which allowed us to rule out primary sclerosing cholangitis (PSC). A second ophthalmological consult did not reveal the presence of a Kayser–Fleischer ring. No significant lesions were detected during the gastroscopy. Laboratory tests showed a decrease in potassium (2.79 mmol/L), vitamin 25 (OH) D3 (8 ng/mL), and total protein (6.36 g/dL), and an increase in fasting glucose (100 mg/dL), total bilirubin (28.13 mg/dL), liver enzymes (ALT—445 U/L, AST—115 U/L), ALP (177 U/L), triglycerides (470 mg/dL), fibrinogen (458 mg/dL), prothrombin time, and INR (PT-13.6 s, INR 1.24). Normal albumin levels were observed. Due to abnormal liver tests and particularly high total bilirubin concentrations, a Sheldon catheter was placed, and plasmapheresis was performed four times. The R ratio was determined and was found to be 7.98, indicating a hepatocellular type of liver injury (20). According to the guidelines presented by the Drug-Induced Liver Injury Network (DILIN), liver injury can be assessed using the R ratio equation [31]: *R* = [$\frac{ALT}{ULN_{ALT}}{]}^{*}$:${\left[\frac{ALP}{ULN_{ALP}}\right]}^{*}$, where ALT—alanine aminotransferase, ALP—alkaline phosphatase, ULNALT—upper limit normal of ALT, ULNALP—upper limit normal of ALP. An asterisk indicates the highest level of total bilirubin. Interpretation of the R ratio is made by reference to a 3-grade scale of the R-value, where <2 means cholestatic liver damage; =2–5 mixed liver injury; >5 hepatocellular liver injury.

Additionally the Roussel Uclaf Causality Assessment Method (RUCAM) scale was used as the diagnostic algorithm for assessing the causality of liver damage. The RUCAM score obtained was six, consistent with probable ashwagandha-induced liver injury. The parameters evaluated by the RUCAM scale for the described patient case are shown in Table 1.

Increased copper excretion was determined in a 24 h urine collection (254 micrograms/24 h). Serum copper levels, as well as ceruloplasmin levels, were within a normal range; however, due to inflammatory processes, ceruloplasmin deficiency may be masked by excessive hepatic release. Penicillamine and ursodeoxycholic treatments were implemented. After 3.5 months of intensive pharmacological treatment, plasmapheresis, and discontinuation of ashwagandha, liver tests and total bilirubin levels normalized. The patient’s general condition improved considerably. Major causes of liver damage were excluded, confirming that herbal supplements containing ashwagandha were most likely the causative agent of drug-induced liver injury (DILI). According to this calculation, in the presented case, liver injury was classified as hepatocellular with pronounced hyperbilirubinemia. After 3.5 months of pharmacological treatment and discontinuation of ashwagandha, liver tests and total bilirubin levels normalized.

## 3. Discussion

A molecular study by Siddiqui et al. [32] concerning the background of ashwagandha-induced liver injury suggests an important role for one of the withanolides—withanone (VIN). It is one of the more representative withanolides of ashwagandha, and is the main metabolite in significant amounts in the extract [33,34]. In addition, and most significantly, VIN consists of electrophilic functional groups associated with adverse drug reactions, referred to as toxicophores, responsible for toxicity [35,36]. It appears that if the endogenous antioxidant defense system against oxidative cell damage, consisting of the most abundant antioxidant, reduced glutathione (GSH), is overloaded due to drug overdose or reduced GSH levels, severe liver damage occurs. The maintenance of reduced glutathione concentrations at a certain level in most cell types emphasizes its vital and multifunctional role in the control of various biological processes, such as detoxification of foreign and endogenous compounds, but also protein folding, regeneration of vitamins C and E, maintenance of mitochondrial function, antiviral defense, regulation of cell proliferation, apoptosis, and immune response. In addition, Win has been shown to form non-labile adducts with the nucleosides dG, dA, and dC, leading to impaired biological properties. Low or reduced GSH levels can affect the detoxification of withathion, contributing to the disruption of DNA transcription, replication, and repair, consequently leading to the disruption of replication, mutagenesis, and apoptosis. This evidence suggests that ashwagandha used in excess stimulates a reduction in GSH levels in cells, which translates into cytotoxicity and may explain the causes of liver damage caused by its consumption. Two types of damage occur most frequently. The first is herbal-induced liver injury (HILI), which is an important and increasingly concerning cause of liver toxicity. The second type is drug-induced liver injury (DILI), which accounts for 10% of all acute hepatitis cases, and thus remains one of the most common causes of acute liver failure [37]. Several cases of DILI after the use of ashwagandha herbal products have been published. Inagaki et al. described a patient taking a single dose of ashwagandha in 2013. In 2014, as an attempt to alleviate high-stress levels, he increased the recommended dose 2–3 times. That same year, he was hospitalized for jaundice. After discontinuation of ashwagandha supplementation, liver parameters normalized. However, after 7 days, the patient’s condition worsened, as confirmed by high values of total bilirubin, AST, ALT, and ALP [38]. In another study, Bjornsson et. al. described the effects of ashwagandha on liver function in five patients. All of them developed jaundice, had elevated liver enzymes, and were diagnosed with drug-induced liver damage, which regressed after eight months [30].

In 2021, Weber et al. presented a case of a 40-year-old man, who after the use of ashwagandha, experienced an acute liver injury. For more than a year, he had consumed ashwagandha extract (500 mg) before he switched to Ashwagandha Now (450 mg, Now Foods, Bloomingdale, IL, USA). After 20 days of consumption, he experienced jaundice, pruritus, and elevation of transaminases [39]. In addition, in 2021, Ireland et al. showed a 39-year-old female who was admitted to the hospital with a one-week history of nausea and jaundice after using an herbal product containing ashwagandha root extract. Initial investigations showed increased levels of liver enzymes. A liver biopsy was performed and showed acute cholestatic hepatitis with confluent necrosis [40]. The presented case of a 23-year-old patient, as well as other documented cases, indicates the need for increased research into the potential role of plant-derived biological compounds. In a report published by Inagaki et al., increasing the dose of a supplement containing *W. somnifera* most likely led to DILI. Therefore, it is extremely important to monitor the dose and frequency of intake of herbal products. Poisoning with unidentified chemicals may be associated with the later onset of symptoms, so it would be appropriate to perform a full diagnostic workup and ensure that the patient receives medical care after symptoms of liver damage have resolved. In each of the reported patients, symptoms of liver injury had a different onset after the administration of herbal drugs. Moreover, after the discontinuation of HDS, the normalization of liver parameters occurred at different time intervals. This may be associated with different forms of liver injury, which have been highlighted in this study. Thus, it appears that further studies focusing on mutation analysis, DNA repair, and protein adduction would help to understand the potential mechanism of the described ashwagandha-induced liver damage. Understanding the molecular mechanisms underlying certain signaling pathways in the metabolism of plant products also seems to be of great importance.

## 4. Conclusions

Identifying the main cause of liver damage in patients taking herbal preparations is extremely difficult, as they are often a mixture of different ingredients in which identifying a single causative factor can be problematic [41]. It appears that legislation should be introduced for the mandatory testing of individual ingredients in herbal and dietary supplements for a large population. This would eliminate the risk of using hazardous substances that could result in serious damage to the body. These measures would significantly improve and shorten the diagnostic process, which, nowadays, is quite lengthy due to the lack of adequate data on substances in HDS (herbal and dietary supplements). Ashwagandha should be more often considered as a potential liver-damaging factor, and doctors should pay attention to herbal supplements taken by patients when collecting anamnesis [30]. A procedure that could optimize hospitalization is a liver biopsy to accurately determine the histopathologic type of liver injury. However, it is unlikely that a liver biopsy would be performed because of the prolonged diagnostic process and the normalization of liver parameters after intensive drug treatment. The role of other substances or drugs, as well as the influence of polymorphisms of genes encoding drug-metabolizing enzymes, are not excluded. At the same time, we would like to emphasize that the inadequate knowledge of the substances contained in HDS mixtures and their mode of action entails diagnostic problems at every step, including treatment. In view of several documented cases of liver damage caused by ashwagandha and the unknown metabolic mechanisms of substances contained in it, attention should be paid to patients reporting the use of these products in the past and presenting symptoms of liver damage.

## Figures and Tables

**Table 1 ijerph-20-03921-t001:** Parameters evaluated by the RUCAM scale for the described patient case.

RUCAM Causality Assessment			
No.	Parameter	Case Described	Score
1.	Time to onset	Patient was hospitalized after 3 months	2 points
2.	Course	Significant decrease in Alk P level after 3.5 months of treatment	2 points
3.	Risk factors	The patient denied using alcohol and he was below 55 years old	0 points
4.	Contaminant drugs	The patient denied taking any other medication	0 points
5.	Exclusion of other causes of liver injury	All of the test were performed and two groups of risk factors were excluded	2 points
6.	Previous information on hepatotoxicity of the drug	No information in medical history	0 points
7.	Response to readministration	Not performed or not interpretable	0 points

## Data Availability

Data sharing is not applicable to this article.

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
