# Peer review of "Liver Dangers of Herbal Products: A Case Report of Ashwagandha-Induced Liver Injury"

_ijerph, 2023, doi:10.3390/ijerph20053921_

Round 1

Reviewer 1 Report

The manuscript “Liver Dangers of Herbal Products–a case report of Ashwagandha induced liver injury”, by Marta Lubarska and coworkers, presents observations on a case of liver injury that could be triggered by the consumption of an Ashwagandha (Withania somnifera) herbal supplement. The medical observations related to this case are intended to be published as a “Case Report”.

For me, a better title is: “Liver Dangers of Herbal Products: A case report of Ashwagandha induced liver injury”

The introduction provides enough information on the context of this report. Probably, more information previously reported (if available) on the effects of known components of the herbal supplement should be included. This could support the author's asseverations about the cause of the liver injury.

The meaning of acronyms must be specified. Although many of the acronyms are well-known by medics, some readers of other fields may have difficulties understanding the text without these meanings.

Considering the importance of this kind of report, it would be interesting the inclusion of some relevant data as Supporting Information. This is a suggestion for authors.

I have included some comments in the PDF document as well.

I consider that this paper can be published in the International Journal of Environmental Research and Public Health after minor revisions.

Reviewer 2 Report

1. In abstract, a comma (,) should be placed after Indian.

2. As per the Indian traditional medicine system (Ayurveda) Ashwagandha should not be consumed by 23-year-old person, then question is why the patient taken Ashwagandha supplement. This should be mentioned.

3. Without knowing the dose and quality of the preparation a conclusion can't be made.

4. Patient should be diagnosed for STD and HIV and patient sexual activity should be recorded.

5. A case report of only one patient can't prove that Herbal Products /Ashwagandha induced liver injury. 

----------------

Reviewer 3 Report

Herbal and dietary supplement (HDS) use has been associated with liver injury, causing concern to the medical community and the public. Lubarska et al present an interesting and rare case of suspected Ashwagandha induced liver injury. The manuscript needs to be re-organized for a better presentation of the case. Specifically:

- Other similar cases should be discussed in the discussion section instead of the introduction. Some reported cases are missing, the authors may consider this review https://www.mdpi.com/2673-4389/2/3/11#B88-livers-02-00011 for adding relative references

- The conclusion section should be shortened and content should be moved to the discussion.

- Please present the lab tests in a table and explain how you determined the R ratio and its significance for the diagnosis of drug induced liver injury (DILI).

- Please highlight your differential diagnosis.

- Please check manuscript for typos and repeated phrases, for example lines 100-101 and 106-107

Reviewer 4 Report

A critical case report on the dangers associated with herbal products, which are widely used by many people worldwide to improve health. Unfortunately, many of these natural products could be hazardous without proper safety testing. The study shows good clinical evidence based on several biomarkers and the effect of treatment on the recovery of the treatment. 

Case raport on top must be “Report”

Line 58: extra space

Line 63: with a without – typo

Line 101: extra space

Line 112: extra space

Line 117: extra space

It would have been helpful if the authors presented information on the timeline of usage of A Ashwagandha and the dosage. Also, the dietary habits of the patient could act as a variable. Another important thing is to look for any diseases that would have caused a similar increase in the biomarkers that the authors saw in the patient. Overall, a well-researched case report with sound methodology and results.

Round 2

Reviewer 2 Report

NA

Author Response

Thank You